# Investigating the dynamics of tax evasion and revenue leakage in somali customs

**Mohamed Ibrahim Nor** [1]*, **Abdinur Ali Mohamed**[2]

**1** Faculty of Management Science, SIMAD University, Mogadishu, Somalia, **2** Faculty of Economics, SIMAD University, Mogadishu, Somalia

* m.ibrahim@simad.edu.so

**Data Availability Statement:** Our data is provided as a sumpplimentary file.

**Funding:** We express our profound gratitude to SIMAD University's Center for Research and Development for their generous funding and

## Abstract

This study aims to investigate the dynamics of tax evasion and revenue leakage in the Somali customs framework, providing insights into the systemic opportunity structures, tax governance deficiencies, and personal incentive structures that facilitate these practices. By applying agency theory and rent-seeking theory, this research seeks to deepen the understanding of the complex relationship between individual motivations and systemic vulnerabilities in exacerbating corruption and tax evasion in a post-conflict governance context. By employing structural equation modeling (SEM) within the ADANCO-SEM analysis framework, this study analyzes primary survey data. This approach allows for a comprehensive examination of the relationships between systemic, governance, and personal factors contributing to corruption and tax evasion. The findings reveal a significant positive relationship between systemic opportunity structures, tax governance deficiencies, and personal incentive structures and the prevalence of tax evasion and corruption. Specifically, systemic opportunity structures were found to significantly influence both tax governance deficiencies and personal incentive structures, highlighting the intertwined nature of these factors in facilitating corrupt practices and tax evasion in Somali customs. This study underscores the urgent need for comprehensive reforms targeting systemic vulnerabilities, enhancing tax governance frameworks, and aligning personal incentives with the public interest. Practical applications include the adoption of advanced technological solutions for improved monitoring and transparency, as well as the development of targeted training programs for customs officials to foster ethical standards and compliance. This research contributes to the existing body of knowledge by providing a unique empirical examination of corruption and tax evasion in the context of Somali customs, a largely underexplored area in the literature. By integrating agency theory and rent-seeking theory, this study offers novel insights into the mechanisms of corruption and tax evasion, highlighting the importance of addressing both systemic and individual factors in combating these issues.

## Introduction

Tax evasion is a global issue affecting economic stability and societal equity [1]. It involves misrepresenting financial affairs to minimize tax liabilities. Examples include Bangladesh's misuse

invaluable technical support throughout the course of this study. Their commitment to fostering research excellence and providing resources has been instrumental in the successful realization of our project. The insights and expertise offered by the team at the Center have significantly contributed to the depth and quality of our work. We are deeply appreciative of their unwavering support and belief in the potential of our study.

**Competing interests:** The authors have declared that no competing interests exist.

of pre-shipment inspection services [2], India's false returns [3], Europe's CumEx-Files scandal [4], the US IRS's detection of tax evasion tactics [5], and African nations' struggles with corporate tax compliance and underreporting income [6]. The challenge lies in establishing effective countermeasures to ensure fiscal integrity and fairness.

The challenge of domestic revenue mobilization in sub-Saharan African countries is a critical concern for sustainable economic development and governance. In the broader context of the region, the necessity of raising internal funds has been emphasized as a pivotal strategy to reduce reliance on foreign aid and to create fiscal space essential for national development [7]. However, the effective mobilization of these revenues is continually undermined by widespread tax evasion and avoidance, coupled with inadequate enforcement of tax laws and prevalent tax corruption [8]. In Somalia, as in many fragile states, the issue of tax evasion is not limited to lost revenue; it significantly affects economic efficiency, income distribution, and the legitimacy of the government [9]. The context in Somalia is particularly complex, with most federal states, except for Puntland, struggling with proper tax collection.

Long-standing challenges stemming from political instability and the presence of insurgent groups, notably Al-Shabaab, have persistently disrupted trade and customs enforcement. This instability has affected the development of critical infrastructure and the application of modern technology in customs systems, hindering efficient trade management and revenue collection. Recognizing the vital role of customs operations in economic sustainability, the Somali government, with international assistance, has been striving to rebuild and modernize these systems. Efforts are underway to enhance the legal and regulatory framework in alignment with global standards, aiming to streamline customs operations and foster economic growth through trade [10]. Additionally, regional collaboration plays a crucial role in addressing cross-border trade challenges and enhancing security. However, the overall effectiveness of these initiatives remains closely tied to the broader context of security and governance within Somalia [11].

The Somali socioeconomic landscape is characterized by conflict, poverty, unemployment, weak governance, and security concerns. These factors contribute to the prevalence of tax evasion and corruption. The economy is heavily reliant on foreign aid and the informal sector, fostering an environment conducive to financial irregularities. Tackling these issues requires a holistic approach, addressing both the symptoms and root causes of the problem. In a recent development, Somalia has issued arrest warrants against high-ranking officials over corruption allegations, signaling a move toward accountability. Despite ongoing reforms in public financial management and achievements in debt relief, corruption in revenue agencies such as customs remains a significant challenge. As Somalia transitions into the post-HIPC era, addressing corruption is imperative for enhancing domestic revenue mobilization and financing critical public expenditures.

Since the reinstatement of the Somali Federal Government (FGS) in 2012, significant reforms have been undertaken in key governmental institutions, notably in public financial management (PFM). These reforms have been instrumental in advancing Somalia's debt relief efforts under the Heavily Indebted Poor Countries (HIPC) initiative. However, despite these achievements, corruption, particularly financial corruption, remains a pervasive issue within government agencies, including revenue collection entities such as customs. Mogadishu International Sea Port, a vital hub in international trade and commerce, is central to Somalia's economic stability. However, custom officials have been accused of engaging in tax evasion and corruption, undermining the integrity of customs operations. These malpractices not only impede effective revenue collection but also erode public trust in the system. As Somalia entered its post-HIPC era, tackling corruption is imperative not only for maintaining integrity but also for economic reasons. Enhancing domestic revenue mobilization is crucial for

financing key public expenditures, including the needs of the armed forces. Despite ongoing efforts to combat these challenges, there is a lack of comprehensive understanding of the factors driving tax evasion and corruption at Somali customs. The complexity of this issue lies in the interplay of institutional, regulatory, and behavioral factors in the customs environment. This study aims to address this gap by employing the principal-agent model and rent-seeking model to analyze the multifaceted dynamics of tax evasion and corruption. The findings are expected to contribute to the international discourse on customs governance in challenging environments and support targeted policy reforms to enhance fiscal integrity and fairness in Somalia. Hence, this study aims to investigate the dynamics of tax evasion and revenue leakage in Somali customs.

## Literature review

### Tax evasion

Tax evasion is a significant issue in modern economies and involves deliberate and illegal attempts by individuals or corporations to reduce their tax liabilities through misrepresentation by tax authorities [1]. This practice includes under-declaring income, exaggerating deductions, and even bribing officials in high-corruption areas. It contributes to the "tax gap", is prevalent in the informal economy and poses a major challenge to fiscal systems worldwide. In 1972, Nobel laureate economists Allingham and Sandmo developed an economic model of tax evasion based on Gary Becker's theory of crime [12]. However, further research indicates that other factors, such as the perceived effectiveness of government expenditures and public participation in decision-making, influence evasion tendencies. Tax evasion is not only an economic issue but also a significant factor in societal inequality, disproportionately affecting the distribution of wealth, with the rich and powerful more likely to engage in and benefit from these practices [13].

### Theoretical framework

The theoretical framework of this study, which focuses on corruption dynamics and tax evasion in Somali customs, is grounded in two pivotal theories [14]: agency theory and rent-seeking theory. These theories provide a comprehensive lens through which the phenomenon of corruption and tax evasion can be analyzed and understood in the Somali context [15].

**Agency theory.** Agency theory is a concept in economics and organizational behavior that explores the relationship between a principal and an agent [16]. In this context, the principal is typically an entity or individual who delegates tasks to an agent to act on their behalf [17]. The theory delves into the challenges arising from the differing interests and information between the principal and the agent [18]. This suggests that the agent, while entrusted with specific responsibilities, may have motivations and information that differ from those of the principal [19]. This misalignment of interests and information, known as the "agency problem," can lead to issues such as moral hazard, where the agent may take risks knowing that the consequences will be borne by the principal [20]. In the realm of taxation and corruption, agency theory is often applied to analyze how government entities delegate tax collection duties to officials (agents) and how the dynamics of this relationship may contribute to problems such as tax evasion and corruption [21].

Agency theory, in the context of corruption dynamics and tax evasion, provides a lens through which to understand the complex relationships between the government (principal) and customs officials (agents) [22]. This theory highlights inherent information asymmetry, where the government, which often lacks complete oversight, must rely on customs officials to execute tax collection effectively. In scenarios where officials prioritize personal gains over the

principal's interests, the risk of tax evasion and corruption increases. Agents may exploit this informational advantage to engage in deceptive practices, hindering the government's ability to ensure accurate revenue collection [23]. Agency theory further emphasizes the importance of designing incentive structures and monitoring mechanisms to align the interests of customs officials with those of the government [24]. Implementing transparent systems and checks can mitigate agency problems, reducing the likelihood of tax evasion and corruption. Understanding the dynamics through this theoretical framework allows policymakers to develop strategies that foster accountability and discourage opportunistic behavior within the customs system, thereby promoting a fair and efficient taxation process [25].

**Rent seeking theory.** Rent-seeking theory provides insights into the dynamics of corruption and tax evasion by focusing on individuals or groups seeking economic gain without creating corresponding value [26]. In the context of the public sector, this theory can be applied to understand how certain entities, including customs officials, might engage in rent-seeking behavior to accumulate wealth through manipulative practices within the taxation system [27]. Rent seeking often involves exploiting loopholes in regulations or engaging in corrupt activities to extract economic rents without contributing to the overall productivity of the economy [28]. In the realm of taxation, Rent-Seeking theory highlights how individuals or groups may invest resources not in productive activities but in efforts to influence tax policies or evade taxes for personal gain [29]. Customs officials, driven by rent-seeking motives, may be inclined to seek bribes or engage in fraudulent practices to maximize their personal wealth at the expense of fair and lawful tax collection. Understanding these rent-seeking dynamics is crucial for designing anticorruption measures and tax policies that deter opportunistic behavior and ensure a more equitable and transparent revenue collection process [30].

Rent-seeking theory is an economic concept that examines how individuals or groups seek to obtain economic rents without contributing to the overall wealth or productivity of society [31]. Economic rent refers to income or gains derived from factors that are not related to productive efforts but arise from the manipulation of economic or political conditions [29]. Rent seeking involves using resources to influence government policies, regulations, or the distribution of wealth to gain advantages that do not result from creating new value [32]. In essence, individuals engaged in rent seeking aim to secure benefits for themselves without contributing to the growth or improvement of the economy [33]. This behavior can take various forms, including lobbying for favorable regulations, seeking subsidies, or engaging in corrupt practices to gain advantages in the marketplace [34]. Rent-seeking theory is often used to analyze how such behavior can distort economic efficiency, create inequality, and contribute to corruption within systems where individuals or groups compete for economic advantages through nonproductive means [35].

## Dynamics of tax evasion and corruption: Global perspectives

Tax evasion is a global phenomenon. In Asia, tax evasion is a complex issue, with cases highlighting the need for robust regulatory frameworks and effective enforcement mechanisms [6]. In Bangladesh, the government relied on pre-shipment inspection (PSI) services provided by agencies such as Société Générale De Surveillance SA and its subsidiary Cotecna to prevent import duty fraud. However, these agencies became facilitators of customs duty evasion, leading to the revocation of Cotecna's certificate in 2008 [2]. In India, tax evasion is manifested through false returns, smuggling, document falsification, and bribery, with penalties ranging from 100% to 300% of the undisclosed income [3]. The Pandora Papers leak in 2021 revealed prominent figures in the United Arab Emirates using offshore accounts for tax evasion [36]. In Europe, the CumEx-Files scandal, involving a network of banks, stock traders,

and attorneys, highlights a major instance of tax evasion, with Germany, France, Italy, Denmark, and Belgium collectively losing an estimated $62.9 billion [4]. This scheme underscores the sophisticated methods employed in evading taxes, particularly in countries with advanced financial systems. Similarly, in Greece, tax evasion is intertwined with corruption, indicating a systemic issue that affects both revenue collection and public trust in institutions [37]. Despite its reputation for high tax compliance, Scandinavia faces challenges, with the wealthiest 0.01% of the population evading a significant portion of its taxes [38]. This discrepancy reflects broader issues of inequality and the inadequacy of traditional tax data to capture the complete financial picture, particularly at the higher end of the income spectrum. In the United Kingdom, efforts to combat tax evasion, including targeting offshore accounts and doubling the prosecutions of tax evaders, reflect a growing recognition of the need for more stringent enforcement [39].

In the United States, tax evasion represents a significant challenge to the fiscal integrity of the federal system, as delineated by U.S. law, which defines it as a deliberate and illegal attempt to evade the payment of taxes imposed by federal law. The Internal Revenue Service (IRS) faces considerable challenges in detecting skimming and non-reporting of income among these groups due to the need for extensive investigations [5]. Particularly notable is the prevalent practice of overstating charitable contributions, with church donations being a common focus [40]. The enormous amount of tax evasion is further underscored by IRS estimates showing a staggering tax gap of $345 billion in 2001, which escalated to $450 billion in 2006 [41]. Moreover, a study indicated that approximately 18 to 19% of total reportable income was not accurately reported in 2008, resulting in an estimated unreported income of approximately $2 trillion and a consequent tax gap ranging from $450 billion to $500 billion [42]. Equally, tax evasion poses a great challenge to African nations, as tax evasion is deeply rooted in the continent's diverse economic and political landscapes [6]. African countries, each with unique fiscal dynamics, grapple with tax evasion as a major impediment to revenue mobilization [43]. For instance, Kenya faces significant challenges in differentiating legal tax avoidance from illegal tax evasion, with a substantial proportion of registered companies not fulfilling their tax obligations, pointing to potential tax evasion practices [6]. Similarly, in Ethiopia, tax evasion manifests through underreporting income, inflating business expenses, and smuggling, necessitating the adoption of more sophisticated tax administration systems and rigorous enforcement strategies [44]. In Rwanda, tax evasion not only impacts revenue collection but also undermines public infrastructure and widens income inequality, necessitating reforms in tax policy and administration [45]. These challenges are reflective of broader issues in many African countries where the thin line between tax avoidance and evasion often blurs, highlighting the need for robust, transparent, and efficient tax systems [6].

Tax evasion in customs and international trade presents a formidable challenge to global economic stability and fairness [46]. The first challenge in combating tax evasion is understanding its forms and prevalence. Underreporting and misclassification of goods are rampant and often involve collusion with customs officials and the use of falsified documents [47]. Multinational corporations exploit loopholes in international trade laws through transfer pricing, manipulating profits to minimize taxes. Smuggling, an age-old problem, continues to undermine legal trade and taxation systems [48]. These methods not only lead to significant revenue losses for countries but also create an uneven playing field in global trade [49]. The impact and response to customs tax evasion vary markedly between developed and developing countries. Despite their sophisticated customs and tax systems, developed countries grapple with the complexity of international trade and advanced evasion techniques. In contrast, developing countries face more daunting challenges [50]. Limited resources, weaker institutional frameworks, and higher corruption levels make them particularly vulnerable to tax evasion. This

disparity highlights the need for a global solution that considers the varying capacities and vulnerabilities of different nations [51]

Addressing this global issue requires a multipronged strategy. Enhanced international cooperation is paramount [52]. Sharing information and best practices can help in identifying and addressing cross-border tax evasion schemes. The role of technology cannot be overstated; AI and big data analytics have the potential to revolutionize the detection of trade anomalies indicative of evasion [53]. However, technology alone is not enough. Capacity building by customs and tax authorities, especially in developing countries, is critical. This includes training, resource allocation, and systemic improvements. Furthermore, robust legal and policy reforms are needed to close loopholes and strengthen enforcement mechanisms. The effective implementation of these solutions requires the active participation of international bodies such as the World Trade Organization (WTO) and the International Monetary Fund (IMF). These organizations can facilitate information sharing, provide technical assistance, and help in drafting and enforcing international agreements. Their role in bridging the gap between developed and developing countries is crucial for a balanced and effective global approach [54].

Corruption in customs and port authorities represents a global challenge with profound implications for international trade and governance [22]. Despite extensive studies, this complex issue persists and is influenced by factors such as the nature of customs transactions, political interference, social norms, and its impact on trade and revenue. Customs transactions are inherently susceptible to corruption due to the discretionary power wielded by officials and frequent interactions with traders. Forms of corruption range from misreporting and theft to various types of bribes, including "speed money" to fast-track processes [55]. This susceptibility underscores the need for stringent oversight and transparency in customs operations. The allure of well-paid positions and rent-seeking opportunities in customs administration makes it a target for political interference [56]. Such interference can manifest in favoritism, harassment of political opponents, and significant fiscal losses. This political dimension of customs corruption necessitates strong institutional independence and mechanisms to insulate customs administration from political manipulation [57].

Corruption in customs is often entrenched in social norms and unwritten rules, which can override formal laws. In many contexts, violating these norms can lead to social sanctions. This cultural aspect of corruption requires a paradigm shift in societal attitudes and a reinforcement of ethical standards within communities and institutions [58]. Corruption at ports and borders, manifesting as collusive and coercive practices, severely impacts shipping costs, trade efficiency, revenue collection, and security [59]. The resulting economic distortions necessitate comprehensive strategies that address both the symptoms and root causes of corruption, ensuring fair and efficient trade practices. The private sector plays a crucial role in combating corruption. Collective actions have shown significant success in reducing corrupt practices. Such collaborations highlight the importance of industry-wide standards and the proactive role of businesses in fostering integrity and transparency. A combination of legal, technical, and social reforms is essential to effectively combat corruption in customs and port authorities [60]. Establishing a robust legal framework, simplifying processes, and leveraging technology, such as data analytics, are critical steps. However, these must be complemented by broader social reforms to address underlying causes, as seen in the successful examples of Rwanda and Georgia [61].

## Empirical studies: Tax evasion and corruption

Corruption and tax evasion in customs and port authorities present multifaceted challenges globally. Recent empirical studies employing diverse methodologies have offered deep

insights into this issue. A noteworthy empirical study examined the relationship between public/political corruption and trade tax revenue on tax evasion across 140 countries from 2008 to 2017 [62]. By utilizing a dynamic two-step system-generalized moment method to address issues such as autocorrelation and heteroskedasticity, the study revealed a significant interaction between the corruption perception index (CPI) and international trade activities [62]. However, surprisingly, this interaction was statistically insignificant in certain groups, including countries with low and high levels of corruption and those with trade surpluses [63]. These findings suggest a sophisticated relationship between corruption and tax evasion, influenced by a nation's corruption level and economic conditions [64].

Perception surveys, another important tool in studying corruption, offer a subjective but valuable perspective. Surveys such as those conducted by Transparency International resulting in the CPI provide rankings of countries based on perceived corruption levels [65]. While not quantifying corruption explicitly, these surveys offer a broader understanding of its prevalence and intensity across different nations. These subjective data complement more objective, quantitative research methods, presenting a fuller picture of global corruption trends [66]. Recent advancements in corruption research include direct estimates of bribes from field observations, 'subtraction' estimates comparing official records against actual receipts, and government corruption audits [67]. These methods provide specific corruption estimates in particular settings but face limitations in offering a comprehensive global view [67]. Nonetheless, they are crucial in understanding the localized nature of corruption and its impact on customs and tax evasion practices [67].

## Somali customs: Historical context

The historical trajectory of Somalia's customs operations presents a vivid illustration of a nation striving to navigate through periods of stability, conflict, and recovery [68]. It explores the evolution of the customs system from the pre-civil war era, through the devastation of civil conflict, to the current efforts at recovery and reform [69]. Prior to the civil war, Somalia's customs operations, though basic, played a vital role in the national economy. The country's strategic position along the Horn of Africa made it a pivotal trade hub [70]. This period underscored the potential of a functional customs system for fostering economic growth. Therefore, the stability and effectiveness of the customs system during this era serve as a testament to its potential to contribute to national revenue and economic stability [71]. The onset of the civil war in 1991 marked a turning point, leading to the collapse of the central government and the disintegration of institutional structures, including customs operations [72]. The resulting chaos catalyzed unregulated trade and smuggling, severely impacting the economy [73]. This period highlights the critical link between political stability and effective customs operations. The inability to enforce customs regulations not only led to a loss of revenue but also exposed the country to various security vulnerabilities. The post-civil war era posed unique challenges: widespread poverty, infrastructural deficits, and persistent conflicts [74]. These conditions hampered the re-establishment of an effective customs system. The entrenchment of illegal trade networks during the civil war further complicated the situation [74]. This scenario underscores the need for comprehensive reforms, not only in the customs system but also in the broader political and economic landscape of Somalia. Acknowledging the critical need for a functional customs system, various Somali governments, with the aid of international organizations, initiated substantial reforms [75].

## Research framework and hypothesis development

In our research framework, we explore the relationship between personal incentives and systemic opportunities as determinants of tax evasion and corruption in Somali customs (see Fig 1). Grounded in the principles of agency theory and rent-seeking theories, our model posits that tax evasion and corruption are not solely the result of individual malfeasance but are also significantly influenced by the structure of the system within which individuals operate. Agency theory illuminates the personal incentive structure by examining how discrepancies between the goals of agents (customs officials) and principals (the government and public they serve) can lead to self-serving behaviors when oversight is insufficient or when misaligned incentives are present. Conversely, rent-seeking theory explains the systematic opportunity structure, highlighting how the political and economic environment creates vulnerabilities that individuals exploit for personal gain, such as through securing exclusive access to resources or decision-making processes. Hence, our hypotheses posit that both individual motivations and systemic opportunities jointly contribute to the prevalence of tax evasion and corruption in Somali customs. This dual-focus approach allows us not only to identify the extent to which personal incentives and systemic opportunities influence corrupt practices but also to propose targeted interventions that address both dimensions to effectively reduce corruption and tax evasion in this context.

The notion that personal incentive structures contribute to increased tax evasion is supported by economic theories on rational behavior and empirical evidence from behavioral economics. According to [12], individuals decide to evade taxes when the expected utility of evading taxes, considering the probability of detection and the severity of penalties, outweighs the utility of compliance. This framework suggests that personal gains, such as increased disposable income from unpaid taxes, serve as a strong motivator for tax evasion. Empirical support for this model is provided by [76], who demonstrate that the probability of tax evasion rises as the potential personal return from evasion increases, evidencing that personal incentive structures significantly influence tax compliance behavior.

Further, the concept of revenue leakage due to personal incentive structures can be traced to the misalignment of individual and organizational goals. When individual incentives are structured purely on short-term gains, disregarding the long-term health of the revenue system, there is a tendency towards behaviors that promote revenue leakage. This is articulated in the work of [77], who found that the more the personal incentives of taxpayers diverged from the collective fiscal needs, the greater was the incidence of revenue leakage. These findings align with agency theory, which posits that misalignment between the principal (government)

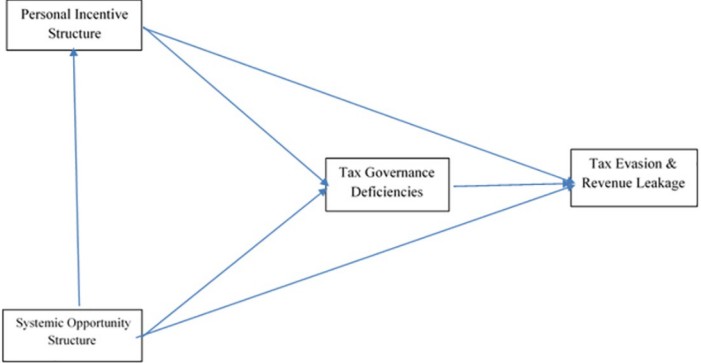

**Fig 1. Research framework.**

and agents (customs officials) can lead to suboptimal outcomes, including diminished revenue collections [78]. Thus, understanding the impact of personal incentive structures is crucial for designing policies that effectively minimize tax evasion and limit revenue leakage. This discussion suggests the following hypotheses:

H1: Personal Incentive Structure Increases Tax Evasion and Revenue Leakage

To substantiate the hypothesis that personal incentive structures increase tax governance deficiencies, one can draw on the literature that explores the interplay between individual motivations and institutional oversight. According to the theory proposed by [79], when personal incentives are not aligned with institutional goals, individuals within a tax administration may engage in practices that undermine governance, such as reducing compliance checks or colluding with taxpayers. This assertion is further supported by empirical research by [80], which suggests that the more significant the misalignment between the personal rewards available to tax officers and the objectives of the tax system, the more pronounced the governance deficiencies are. These findings highlight the critical role of incentive structures in shaping behavior within tax authorities, thereby influencing the overall effectiveness of tax governance.

Moreover, the broader impacts of personal incentive structures on tax governance can be seen in the form of reduced transparency and accountability within tax systems. As demonstrated by [81], incentive-driven behavior often leads to less transparent operational practices among tax officials, who may prioritize personal gains over the public good. This behavior typically results in decreased levels of taxpayer trust and compliance, further exacerbating governance challenges. The study by James and Alley illustrates how personal incentives can lead to a cycle of decreasing transparency and increasing tax evasion, which ultimately undermines the capacity of the tax system to function effectively. Thus, the design of incentive structures within tax administrations must carefully balance individual motivations with the imperative of maintaining robust and transparent governance mechanisms. This discussion suggests the following hypotheses:

H2: Personal incentive structure increases tax governance deficiencies.

The hypothesis that tax governance deficiencies increase tax evasion and revenue leakage is well-supported by the literature on public economics and institutional effectiveness. [82] argue that when tax governance is weak—characterized by inadequate regulatory frameworks, insufficient enforcement mechanisms, and lack of administrative capacity—there is a marked increase in tax evasion practices. This relationship is primarily due to the diminished risk of detection and the consequent low cost of non-compliance perceived by taxpayers. The empirical study by [64, 83] reinforces this perspective, showing that countries with weak tax governance structures experience higher rates of tax evasion and significant revenue losses. These studies collectively suggest that strengthening governance systems can reduce tax evasion, thus enhancing revenue collection and overall fiscal health.

Furthermore, the impact of governance deficiencies on revenue leakage extends beyond mere tax evasion. According to studies by [84–86], weak governance often results in corrupt practices and informal agreements between taxpayers and tax officers, which facilitate tax evasion and avoidance. This erosion of tax integrity leads not only to immediate revenue loss but also to long-term detrimental effects on the tax base, as taxpayers lose confidence in the equity and efficiency of the tax system. The cascading effect of these governance deficiencies highlights the critical need for robust, transparent, and accountable tax administration systems to ensure compliance and minimize revenue leakage. These findings underscore the direct linkage between tax governance quality and the financial integrity of governmental operations,

suggesting that improvements in governance could substantially reduce tax-related malfeasance. This discussion suggests the following hypotheses:

H3: Tax Governance Deficiencies Increase Tax Evasion and Revenue Leakage

The hypothesis that systemic opportunity structures increase tax evasion and revenue leakage is supported by extensive research that examines the impact of the broader economic and regulatory environment on individual and corporate behavior. [86, 87] posit that when the systemic structure offers opportunities such as loopholes, ambiguous tax codes, and weak enforcement mechanisms, it inherently encourages both individual taxpayers and businesses to engage in tax evasion. This is because such structures reduce the perceived probability of detection and the associated costs of evasion, making non-compliance a more attractive option. Similarly, research by [88, 89] corroborates these findings, indicating that the more permissive the system is, due to complex or poorly implemented tax laws, the greater the extent of tax evasion and the resultant revenue leakage observed.

Moreover, the concept of systemic opportunity structures extends to the socio-economic dimensions that influence tax behavior. Studies by [90, 91] highlight how systemic inequalities and perceived injustices in the tax system can lead to increased tax evasion. For instance, if certain segments of the population or business sectors benefit disproportionately from tax exemptions or credits, this can create a sense of unfairness and motivate other groups to evade taxes, thus contributing to broader revenue leakage. This viewpoint is supported by [92], who argue that the perception of inequality in tax treatment can undermine voluntary compliance and erode the tax base. Therefore, addressing these systemic discrepancies is crucial for reducing opportunities for tax evasion and securing the integrity of revenue systems. Therefore, this study hypothesizes the following:

H4: Systemic Opportunity Structure Increases Tax Evasion and Revenue Leakage

The hypothesis that systemic opportunity structures increase tax governance deficiencies is grounded in the premise that broader economic and administrative frameworks can significantly affect the efficacy of tax systems. According to studies by [93, 94], systemic flaws such as complex tax regulations, decentralized administrative responsibilities, and lack of clear accountability mechanisms often lead to inefficiencies and vulnerabilities in tax governance. These structural complexities can obscure oversight, making it difficult to enforce tax compliance effectively and thus fostering governance deficiencies. Furthermore, as highlighted by [95, 96], when the tax system is characterized by high levels of bureaucratic discretion and inadequate regulatory oversight, it provides fertile ground for corruption and inefficiencies, which further degrade the governance quality.

In addition to the administrative and regulatory aspects, the economic context of a tax system also plays a crucial role in shaping governance outcomes. [97, 98] argue that economic environments marked by high tax rates and economic disparity often encourage tax authorities and taxpayers to engage in collusive behaviors to circumvent tax liabilities, thus exacerbating governance deficiencies. This is particularly evident in contexts where the opportunity costs of compliance are perceived to be high compared to the benefits of evasion. The interplay between these economic incentives and systemic structures underscores how deeply embedded opportunity structures can weaken tax governance, leading to increased instances of evasion and significant revenue leakage. This complex relationship suggests that reforms aimed at simplifying tax laws and enhancing transparency and accountability in tax administration are essential for mitigating governance deficiencies and improving overall tax compliance. Therefore, this study hypothesizes the following:

H5: Systemic Opportunity Structure Increases Tax Governance Deficiencies

The hypothesis that a systemic opportunity structure increases the personal incentive structure for tax evasion is strongly supported by the literature on the interdependence of system-level characteristics and individual decision-making in economics and behavioral sciences. The framework developed by [99] on the economic theory of crime suggests that individuals respond rationally to the incentives provided by the external environment, including the legal and economic frameworks. If the system provides loopholes, low penalties for non-compliance, and a high potential for undetected evasion, it naturally enhances the personal incentives for individuals and corporations to engage in tax evasion. This viewpoint is substantiated by the work of [46, 100], who found that perceived weaknesses in the tax system, such as the complexity of tax laws and the inefficiency of tax authorities, directly encourage taxpayers to exploit these vulnerabilities, thus aligning their personal incentives with the opportunities provided by the system.

Furthermore, the interaction between systemic opportunities and personal incentives is also highlighted in the study of social norms and their impact on tax compliance, as discussed by [101]. They argue that when systemic structures are perceived as unfair or overly complex, it can lead to a deterioration of social norms regarding tax compliance. This shift in norms adjusts personal incentives, making tax evasion more socially acceptable and personally beneficial. The empirical evidence provided by [102] supports this, demonstrating that in systems with widespread opportunities for evasion, individuals are more likely to rationalize their non-compliant behavior as justified or harmless, thus increasing their personal incentive to evade taxes. These studies collectively illustrate how systemic opportunity structures not only provide the means but also influence the motivational drivers behind tax evasion, thereby enhancing the personal incentive structure for such behaviors. Therefore, this study hypothesizes the following:

H6: A systemic opportunity structure increases the personal incentive structure.

## Materials and methods

### Research design

This study adopts a comprehensive and systematic approach to investigate the dynamics of tax evasion and revenue leakage in Somali Customs. By integrating a quantitative research framework, the design ensures a structured and empirically grounded exploration of these critical issues. At the heart of this approach is the implementation of structural equation modeling (SEM), a sophisticated analytical method that surpasses the capabilities of traditional multiple regression analysis. SEM stands out for its ability to assess complex variable relationships and latent constructs that underlie the observable phenomena of corruption and tax evasion. This method is particularly suited for testing the theoretical frameworks of the principal-agent and rent-seeking models in the context of Somali customs. Furthermore, SEM's flexibility in model specification and its capacity to handle multiple equations simultaneously provide a robust platform for examining the interdependencies among variables. This approach is instrumental in capturing the complex dynamics of tax evasion and revenue leakage, offering insights into the latent variables that drive these phenomena. Through SEM, this study aims to construct a holistic model that encapsulates the multifarious nature of corruption and tax evasion in Somali Customs, thereby providing a richer, more nuanced analysis. The application of SEM in this research design is poised to offer significant contributions to the discourse on tax evasion and revenue leakage, particularly in contexts similar to Somali customs. By leveraging the

advanced capabilities of SEM, this study is expected to uncover deeper insights and more complex interactions among the variables of interest, thus informing more effective policy interventions and reform strategies. The employment of SEM in this study represents a methodological advancement, enhancing the rigor and depth of the investigation into the dynamics of tax evasion and revenue leakage.

### Data collection and sampling method

In this study on tax evasion and revenue leakage in Somali Customs, a diligently designed data collection and sampling strategy plays a crucial role. The primary empirical data are sourced from a survey conducted in August 2023, which offers a structured and effective means of gathering information. This survey method, notable for its thoroughness, targets a diverse group of participants from various sectors of Somali society, including the government, the private sector, and the social sector, encompassing NGOs and INGOs. This study used a purposive or stratified sampling method to select participants from various sectors within Somali society. Such a purposive or stratified sampling approach is pivotal for the study, as it aligns seamlessly with the research objectives, allowing for an in-depth exploration of the multifaceted nature of corruption and tax evasion. By incorporating the perspectives of key stakeholders and ensuring a representative cross-section of society, this study is able to delve into the intricacies of these issues across different contexts and among diverse groups. This approach not only acknowledges the complexity of corruption dynamics but also ensures that the findings reflect the interplay of various actors within Somali society. Of the 150 questionnaires distributed, 97 passed the stringent standards of data cleaning and cleansing, providing a solid empirical foundation for further analysis. This rigorous process underscores the study's commitment to inclusivity, comprehensive examination, and its dedication to delivering a holistic and insightful analysis of the pressing issues at hand.

### Construct development and variable measurements

In our study, construct development and variable measurements are crafted to dissect the intricate dynamics between personal incentives and systemic opportunities as drivers of tax evasion and corruption in Somali customs. For personal incentives, we explore psychological and moral factors and motivations for tax evasion. These aspects are quantified using psychometric scales to capture the attitudes and beliefs of the various stakeholders toward tax evasion and corruption. Systemic opportunities are encapsulated through external environmental factors and institutional factors, examining variables such as regulatory quality, institutional transparency, and the ease of exploiting system vulnerabilities for personal gain. By operationalizing these constructs through such targeted variables, our research framework aims to illuminate the interplay between individual motivations and environmental conditions that underpin tax evasion and corruption, thereby offering a comprehensive analysis grounded in agency and rent-seeking theoretical perspectives (see Table 1).

### Results

First, utilizing SPSS, we identified missing values in our dataset and subsequently addressed these gaps through the application of linear interpolation to ensure data completeness and integrity. Second, we checked common method bias in the dataset. The proactive step to evaluate common method variance (CMV) in data collected from a single source, as underscored by Podsakoff, MacKenzie [103], through the implementation of Harman's single factor test, represents a critical methodological safeguard in the pursuit of robust research findings. This adherence to established protocols by incorporating all principal constructs into a principal

**Table 1. Construct development.**

| No. | Variable (construct) | Definition | Survey items (questions) |
|---|---|---|---|
| 1 | Tax Evasion and Revenue Leakage (TERL) | The processes and outcomes where individuals or entities illegally avoid paying taxes, leading to a significant loss of government revenue. | Q8: Customs officer replaces high-value items with lower-tariff items |
| | | | Q9: Tampering with the bill of lading and packing list in collaboration with customs officers |
| | | | Q10: Customs officers encourage traders to avoid official channels |
| | | | Q11: Collaboration among government agencies to facilitate tax evasion |
| | | | Q48: The Government loses 50%—70% of the actual customs revenue. |
| 2 | Tax Governance Deficiencies (TGD) | The shortcomings and inefficiencies in a country's tax administration and policy frameworks that facilitate tax evasion and contribute to revenue leakage. | Q12: Businesspeople engage in bribery to save or gain money |
| | | | Q13: Businesspeople devise innovative methods to persuade tax officers to participate in tax evasion |
| | | | Q14: Inadequate monitoring facilitates tax evasion and corruption |
| | | | Q15: Weak penalties and sanctions encourage noncompliance |
| | | | Q16: Insufficient resources hinder effective enforcement efforts |
| | | | Q17: Limited coordination among relevant authorities contributes to misconduct |
| | | | Q18: Lack of transparency fosters an environment conducive to corruption |
| **Agency theory: Personal Incentives Structure** | | | |
| 3 | Personal Incentive Structure (PIS) | The individual motivations, benefits, and deterrents that drive a person's decision to comply with or evade tax obligations, directly impacting the extent of revenue leakage | Q3: Lack of transparency and accountability |
| | | | Q4: Weak regulatory enforcement |
| | | | Q5: Insufficient penalties and deterrents |
| | | | Q6: Collusion between officials and businesses |
| | | | Q7: Inadequate knowledge and awareness of tax obligations |
| | | | Q28: Perception of low risk and high reward |
| | | | Q29: Desire to minimize tax obligations |
| | | | Q30: Lack of trust in government institutions |
| | | | Q31: Perception of widespread corruption |
| | | | Q32: Greed and personal gain |
| **Rent-Seeking theory: Systemic Opportunities Structure** | | | |
| 4 | Systematic Opportunity Structure (SOS) | The regulatory, legal, and institutional frameworks that either facilitate or hinder the occurrence of tax evasion and the subsequent loss of government revenue. | Q35: Pressure on tax havens |
| | | | Q38: Enhancing transparency and accountability measures |
| | | | Q39: Improving tax education and awareness |
| | | | Q40: Increasing resources for oversight and monitoring |
| | | | Q41: Enhancing coordination among relevant authorities |
| | | | Q42: Automation of tax processes |
| | | | Q43: Digitization of tax records and transactions |
| | | | Q44: Enhancing data analytics and risk assessment capabilities |
| | | | Q45: Implementing secure and efficient online payment systems |
| | | | Q46: Strengthening cybersecurity measures. |

component factor analysis, following the methodology recommended by [104], is not merely a procedural formality but a foundational effort to ensure the integrity of the study's conclusions. The finding that the first three factors account for 81.47% of the variance not only surpasses the threshold indicative of mitigated common method bias concerns but also indicates a broad capture of the multifaceted essence inherent within the dataset. This significant dispersion suggests that the study's findings are anchored in a rich tapestry of underlying constructs rather than being unduly influenced by the singular dimensionality often feared in single-source data analyses. Consequently, this analytical rigor bolsters the argument that common method bias has been effectively addressed and minimized, affirming the study's empirical credibility and the reliability of its insights (see, for instance, [103, 105, 106].

In this investigation, ADANCO-SEM analysis was employed to analyze the data [107, 108]. ADANCO is a software application for composite-based structural equation modeling (SEM), a statistical technique used for analyzing complex data relationships between measured variables and latent constructs [104, 105]. The analysis involved a thorough evaluation of the measurement model to ascertain its validity and reliability, as well as an assessment of the structural model to test the relationships among variables using the bootstrapping method [109].

## Measurement model

In the process of assessing the measurement model, this study evaluated both convergent validity and discriminant validity, adhering to the guidelines proposed by [109]. Convergent validity was assessed through a rigorous analysis of factor loadings, average variance extracted (AVE), and composite reliability (CR). The empirical findings from this evaluation are presented in Table 2, which highlights that all the specified criteria have been satisfactorily met. Specifically, the factor loadings for all items surpassed the threshold of 0.7, the AVE values exceeded the benchmark of 0.5, and the CR values were consistently above 0.7. These results collectively affirm the adequacy of convergent validity for the scale's measurement, thereby underscoring the robustness and reliability of the measurement model in capturing the constructs of interest with precision and accuracy.

To evaluate the discriminant validity of the measurement model, this study employed the heterotrait-monotrait (HTMT) ratio of correlations, an approach grounded in the multitrait-multimethod matrix delineated by [110]. This method hinges on a critical benchmark: the HTMT value must not exceed the HTMT.90 threshold of 0.90, as established by [111], to ensure discriminant validity. An examination of the results, as detailed in Table 3, reveals that all measured values fall below this prescribed threshold, with the sole exception of the TGD. This finding suggests that, across the board, discriminant validity is generally upheld within the model, signaling that the constructs are sufficiently distinct from one another. This outcome bolsters the structural integrity of the measurement model, affirming that it effectively captures discrete phenomena without undue overlap, thus lending credibility to the subsequent analyses and interpretations derived from the model.

## Structural model

This study assessed the structural model's robustness by employing a bootstrapping procedure with a substantial resample of 5,000, as recommended by [109], to ascertain the model's $R^2$, standard beta (β), t values, and effect sizes ($f^2$). Such comprehensive analysis was pivotal for evaluating the theoretical framework's empirical validity. The analysis shown in Table 4 and Fig 2 revealed explicit support for all six posited hypotheses, indicating a significant positive relationship between Systemic Opportunity Structure (β = 0.8260, p < 0.01), Tax Governance Deficiencies (β = 0.9104, p < 0.01), and Personal Incentive Structure (β = 0.8693, p < 0.01)

**Table 2. Convergent validity.**

| No. | Variable | Item | Loading | CR | AVE |
|---|---|---|---|---|---|
| 1 | Tax Evasion and Revenue Leakage (TERL) | TERL1 | 0.9166 | 0.9374 | 0.7552 |
| | | TERL2 | 0.8933 | | |
| | | TERL3 | 0.8596 | | |
| | | TERL4 | 0.9035 | | |
| | | TERL5 | 0.7633 | | |
| 2 | Tax Governance Deficiencies (TGD) | TGD1 | 0.9323 | 0.9652 | 0.7992 |
| | | TGD2 | 0.8455 | | |
| | | TGD3 | 0.9066 | | |
| | | TGD4 | 0.8873 | | |
| | | TGD5 | 0.9004 | | |
| | | TGD6 | 0.8667 | | |
| | | TGD7 | 0.9161 | | |
| 3 | Systematic Opportunity Structure (SOS) | SOS1 | 0.9248 | 0.9841 | 0.8609 |
| | | SOS2 | 0.9109 | | |
| | | SOS3 | 0.9306 | | |
| | | SOS4 | 0.9328 | | |
| | | SOS5 | 0.9255 | | |
| | | SOS6 | 0.925 | | |
| | | SOS7 | 0.9263 | | |
| | | SOS8 | 0.9624 | | |
| | | SOS9 | 0.9412 | | |
| | | SOS10 | 0.8978 | | |
| 4 | Personal Incentive Structure (PIS) | PIS1 | 0.9152 | 0.9673 | 0.7506 |
| | | PIS2 | 0.9353 | | |
| | | PIS3 | 0.927 | | |
| | | PIS4 | 0.8936 | | |
| | | PIS5 | 0.746 | | |
| | | PIS6 | 0.8533 | | |
| | | PIS7 | 0.7189 | | |
| | | PIS8 | 0.7859 | | |
| | | PIS9 | 0.9418 | | |
| | | PIS10 | 0.9109 | | |

and Tax Evasion & Revenue Leakage. Furthermore, it was found that the Systemic Opportunity Structure substantially influences both Tax Governance Deficiencies ($\beta = 0.8156$, $p < 0.01$) and Personal Incentive Structure ($\beta = 0.8398$, $p < 0.01$), with Personal Incentive Structure also positively affecting Tax Governance Deficiencies ($\beta = 0.9158$, $p < 0.01$). These findings not only validate the proposed hypotheses but also underscore the intertwined nature

**Table 3. HTMT ratio.**

| No. | Construct | 1 | 2 | 3 | 4 |
|---|---|---|---|---|---|
| 1 | Tax Evasion and Revenue Leakage (TERL) | | | | |
| 2 | Tax Governance Deficiencies (TGD) | 0.9106 | | | |
| 3 | Systematic Opportunity Structure (SOS) | 0.8296 | 0.8152 | | |
| 4 | Personal Incentive Structure (PIS) | 0.8675 | 0.9137 | 0.8386 | |

**Table 4. Structural model.**

| Hypothesis | Path relationship | Std. beta | SE | t value | Decision | f2 | VIF |
|---|---|---|---|---|---|---|---|
| H1 | Systemic Opportunity Structure →Tax Evasion & Revenue Leakage | 0.8260 | 0.0601 | 13.7493 | Supported | 0.0945 | 9.776 |
| H2 | Systemic Opportunity Structure → Tax Governance Deficiencies | 0.8156 | 0.0677 | 12.0413 | Supported | 0.0478 | 9.776 |
| H3 | Systemic Opportunity Structure → Personal Incentive Structure | 0.8398 | 0.0695 | 12.0763 | Supported | 2.3936 | 9.776 |
| H4 | Tax Governance Deficiencies → Tax Evasion & Revenue Leakage | 0.9104 | 0.0268 | 33.9477 | Supported | 0.4276 | 5.933 |
| H5 | Personal Incentive Structure → Tax Evasion & Revenue Leakage | 0.8693 | 0.0374 | 23.2328 | Supported | 0.0077 | 6.002 |
| H6 | Personal Incentive Structure → Tax Governance Deficiencies | 0.9158 | 0.0337 | 27.1612 | Supported | 1.1731 | 6.002 |

of systemic, governance, and personal factors in exacerbating tax evasion and revenue leakage issues.

In the exploration of tax evasion and revenue leakage and tax governance deficiencies, our study demonstrates compelling findings, with $R^2$ values of 0.851 and 0.846, respectively, indicating that a substantial proportion of the variance was explained by our model. Adhering to the methodological guidance offered by [109], we assessed the effect sizes ($f^2$) to understand the magnitude of the relationships postulated in our hypotheses. According to Cohen, benchmarks for interpreting effect sizes—0.02, 0.15, and 0.35 denoting weak, moderate, and strong effects, respectively—our analysis revealed that all supported hypotheses manifest acceptable effect sizes, thereby affirming the significance and practical relevance of our findings (see Table 4). Moreover, an examination for multicollinearity was conducted, adhering to the variance inflation factor (VIF) threshold of 10.00 [112, 113]. The results reassuringly indicated no concerns of multicollinearity among the variables, thereby underscoring the reliability of our model's estimations and the robustness of our theoretical framework in clarifying the dynamics of tax evasion and governance deficiencies.

## Discussion

The aim of this study is to investigate the dynamics of tax evasion and revenue leakage in Somalia customs. Structural equation modeling (SEM) (the ADANCO-SEM analysis) was used to analyze the data. The analysis revealed a significant positive relationship between Systemic Opportunity Structure, Tax Governance Deficiencies, and Personal Incentive Structure and Tax Evasion and Revenue Leakage. Furthermore, it was found that the Systemic

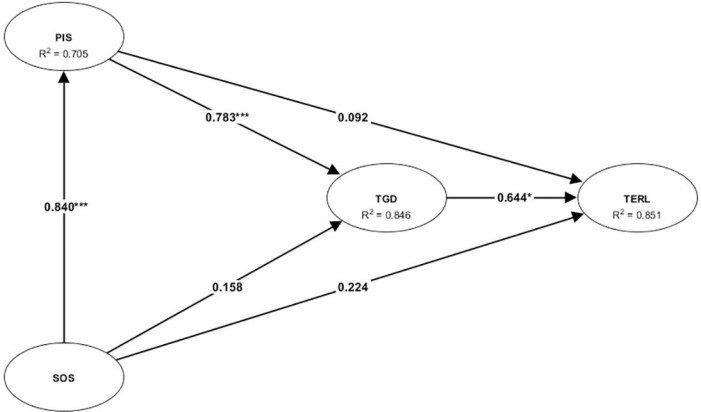

**Fig 2. The results of SEM direct effects inference.**

Opportunity Structure substantially influences both Tax Governance Deficiencies and Personal Incentive Structure. Finally, Personal Incentive Structure is also found to positively affect Tax Governance Deficiencies. These findings underscore the intertwined nature of systemic, governance, and personal factors in exacerbating tax evasion and revenue leakage issues in Somali customs.

The comparative analysis of the study reveals a distinct landscape of tax evasion and revenue leakage in Somali Customs when compared with global practices. Unlike more stable jurisdictions where tax governance deficiencies might be buffered by strong legal systems and advanced technological enforcement mechanisms, Somalia's context is markedly defined by the confluence of systemic opportunity structures and personal incentive structures that directly bolster tax evasion and revenue leakage. This contrast underscores the critical need for Somalia to adopt context-specific measures that address the unique systemic and personal drivers of tax evasion, highlighting a departure from conventional strategies employed in more stable governance contexts.

The Systemic Opportunity Structure acts as a double-edged sword, facilitating tax evasion through gaps in the economic, legal, and social framework while offering avenues for reform. Tax governance deficiencies reflect the critical weaknesses in policy, enforcement, and administrative practices that are exploited for evasion. Meanwhile, the Personal Incentive Structure encapsulates the motivations driving individuals to engage in or resist tax evasion, highlighting the role of perceived fairness and personal benefit. Together, these variables present a comprehensive model for analyzing and addressing the complexities of tax evasion in Somalia. This study illuminates the profound impact of contextual factors on tax evasion and revenue leakage in Somali Customs. The transitional governance structures, post-conflict economic rebuilding, and prevailing socioeconomic conditions create a unique environment that exacerbates the challenges of tax collection and enforcement. These factors not only influence systemic opportunity structures but also shape personal incentives and governance deficiencies, underscoring the importance of tailoring reform efforts to the Somali context to effectively mitigate tax evasion and revenue leakage.

This study's findings resonate with the literature on tax evasion and governance, confirming the complex interplay between systemic structures, governance deficiencies, and personal motivations (see, for instance, [98, 114–117]. The findings corroborate the theoretical framework proposed by [118], which suggests that the complexities of tax systems and the socioeconomic environment significantly affect individuals' propensity to evade taxes. By aligning with these seminal works, the study not only reinforces the established theoretical underpinnings but also extends the discourse by contextualizing these dynamics within a fragile and transitional governance setting, thereby enriching the academic conversation with complex insights into the Somali context (see, for instance, [76, 77, 119]).

This study investigates the dynamics of tax evasion and revenue leakage in Somalia customs, utilizing agency theory and rent-seeking theory as its theoretical backbone. Agency theory unravels the complexities inherent in the relationship between the government and customs officials, revealing the challenges posed by information asymmetry and conflicting interests. Concurrently, rent-seeking theory offers a lens through which to scrutinize the motivations behind engaging in economic gains through nonproductive means, particularly within the taxation system. Together, these theories illuminate the multifaceted nature of corruption and tax evasion, providing a robust framework for analyzing the intricate web of factors that facilitate these practices in the context of Somalia's governance structures. This dual-theoretical approach not only enhances our understanding of the underlying mechanisms of tax evasion and corruption but also underscores the importance of targeted reforms aimed at mitigating these pervasive issues.

The study's application of agency theory to the dynamics of tax evasion and corruption in Somali customs reveals the critical issues stemming from information asymmetry and divergent interests between the government (principal) and customs officials (agents). This theoretical perspective sheds light on the inherent challenges in aligning the objectives of customs officials with those of the government, especially in environments characterized by weak institutional controls and high corruption risk. The findings underscore the significance of implementing robust monitoring mechanisms and incentive structures to mitigate the "agency problem," suggesting that customs officials may engage in corrupt practices or tax evasion when their personal interests diverge significantly from those of the government. By highlighting the complexities of this principal-agent relationship, this study advocates for strategies that not only enhance transparency and accountability within the customs administration but also realign agents' incentives with the overarching goals of efficient tax collection and corruption reduction. This finding has profound theoretical implications, reinforcing the necessity of addressing the underlying factors that contribute to information asymmetry and interest misalignment to effectively curb tax evasion and corruption.

Employing rent-seeking theory, this study delves into the motivations behind individuals' or groups' pursuit of economic gains through manipulation or exploitation of the taxation system without contributing to societal wealth or productivity. This theory illuminates how customs officials may be inclined to engage in rent-seeking behaviors, such as soliciting bribes or facilitating tax evasion, to maximize personal wealth at the nation's expense. The implications of these findings are twofold. First, they highlight the pervasiveness of rent-seeking activities within the customs framework, underscoring the urgent need for reforms that close loopholes and reduce opportunities for corrupt practices. Second, the study's insights point toward the theoretical necessity of crafting policies that discourage rent-seeking by ensuring that wealth accumulation through the public sector demands genuine value addition to the economy. This theoretical perspective advocates for a systemic overhaul that minimizes rent-seeking opportunities, thereby promoting a fairer, more transparent, and productive economic environment. Together, the agency and rent-seeking theories provide a comprehensive framework for understanding the multifaceted nature of tax evasion and revenue leakage in Somali customs, offering critical insights into the development of effective countermeasures.

The findings of this study have significant implications for policy and practice, suggesting an urgent need for comprehensive reforms in Somali customs. Policies should aim to strengthen the tax governance framework by addressing systemic vulnerabilities, enhancing enforcement capabilities, and redesigning tax policies to mitigate evasion opportunities. Practices must also consider individual motivations, advocating for initiatives that promote fairness and compliance. By addressing both systemic and personal drivers of tax evasion, Somali authorities can foster a more resilient and effective tax collection system.

The novel contribution of this study lies in its comprehensive examination of the dynamics of tax evasion and revenue leakage in the context of Somali Customs, employing ADANCO-SEM analysis. By integrating systemic, governance, and personal factors, this study offers a unique perspective on the challenges and drivers of tax evasion in a post-conflict, governance setting. This approach not only enriches the academic discourse but also provides actionable insights for policymakers grappling with similar issues in fragile states. This study significantly augments the literature on tax evasion and revenue leakage by providing empirical evidence from the under-researched context of Somali Customs. By dissecting the interrelations between systemic opportunity structures, tax governance deficiencies, and personal incentive structures, this study extends the understanding of how these factors collectively contribute to tax evasion. Moreover, the study's insights into the specific challenges faced by Somalia offer

valuable lessons for other countries with similar sociopolitical landscapes, thereby broadening the applicability and relevance of its findings.

## Conclusion

This study systematically explores the dynamics of tax evasion and revenue leakage in the Somalia customs framework, yielding several pivotal findings. First, it establishes a significant positive relationship between systemic opportunity structures, tax governance deficiencies, and personal incentive structures and their collective impact on tax evasion and revenue leakage. Moreover, the analysis reveals that systemic opportunity structures significantly influence both tax governance deficiencies and personal incentive structures, while personal incentive structures additionally affect tax governance deficiencies. These results underscore the complex interplay of systemic, governance, and personal factors in exacerbating tax evasion and revenue leakage issues, highlighting the critical need for targeted interventions. By integrating agency theory and rent-seeking theory, this study makes significant theoretical contributions to the literature on corruption and tax evasion. It extends agency theory by illustrating how the principal-agent relationship in Somali customs facilitates corruption and tax evasion, particularly under conditions of information asymmetry and conflicting interests. Concurrently, the application of Rent-Seeking Theory provides insights into how individuals exploit the taxation system for personal gain without contributing to societal wealth, offering a deeper understanding of the motivations behind corrupt practices. These theoretical insights not only enrich the academic discourse but also offer a complex framework for examining corruption and tax evasion in fragile governance contexts.

The findings of this study have profound implications for policymakers, emphasizing the need for comprehensive reforms in Somali customs. To mitigate tax evasion and corruption, policies must address systemic vulnerabilities, enhance tax governance frameworks, and realign personal incentives with public interests. Specifically, the study suggests the development of transparent systems and robust monitoring mechanisms to curb agency problems and discourage rent-seeking behaviors. Such policy measures are crucial for fostering a fair, efficient, and transparent tax collection system, ultimately contributing to economic stability and governance quality in Somalia. In terms of practical applications, this research offers actionable strategies for customs administrators and government officials. Implementing structured training programs to educate customs officials about ethical standards and the long-term benefits of fair tax collection practices can mitigate personal incentives for corruption. Additionally, adopting advanced technological solutions to enhance oversight and reduce informational asymmetries can further deter tax evasion. These practical applications underscore the importance of a multifaceted approach to reform, combining education, technology, and policy to address the root causes of corruption and tax evasion in Somali customs.

The overall significance of this study lies in its comprehensive examination of the interlocking factors contributing to corruption and tax evasion in Somali customs, offering a unique lens through the application of agency theory and rent-seeking theory. By revealing the systemic, governance, and personal dynamics at play, this research provides a critical foundation for understanding the challenges and opportunities in combating tax evasion and corruption. This contribution is invaluable not only for its academic value but also for its potential to guide effective policy and practical interventions in fragile governance settings.

This study, while providing valuable insights into the dynamics of corruption and revenue leakage in Somali customs, has several limitations that merit attention. First, its generalizability is constrained by its singular focus on Somalia, a country with unique sociopolitical and economic conditions stemming from its post-conflict and transitional governance status. Such

specificity may limit the direct applicability of the findings to other contexts with differing governance structures, economic environments, or cultural norms. Furthermore, the complexity of corruption and tax evasion dynamics presents an analytical challenge, as the multifaceted interplay of individual behaviors, systemic structures, and governance mechanisms may not be fully captured within the scope of this study. This leaves potential influencing factors and their intricate interactions unexplored. Additionally, the exclusive reliance on agency theory and rent-seeking theory, while providing a structured theoretical framework, may overlook other theoretical perspectives that could offer deeper or complementary understandings of tax evasion and corruption phenomena. Finally, the predominance of quantitative methods, such as structural equation modeling, in analyzing the data may constrain the depth of understanding regarding the motivations and perceptions of individuals involved in these practices. Incorporating qualitative insights could enrich the study, offering a more nuanced perspective on the underlying causes and motivations of corruption and tax evasion.

Addressing the limitations highlighted in this study opens several avenues for future research that could substantially enrich the academic discourse on corruption and tax evasion, particularly within fragile governance contexts. Future studies could endeavor to broaden the generalizability of findings by incorporating comparative analyses across diverse geopolitical environments, thereby illuminating the influence of varying governance structures, economic conditions, and cultural norms on tax evasion and corruption. Additionally, adopting a more holistic approach that integrates both qualitative and quantitative methodologies could provide deeper insights into the motivations, perceptions, and behaviors of individuals involved in these practices, offering a more nuanced understanding of the underlying dynamics. Expanding the theoretical framework beyond agency theory and rent-seeking theory to include other relevant perspectives could also uncover alternative explanations and complement existing understandings of the phenomenon. Furthermore, exploring the complex interplay of additional influencing factors and their interactions could shed light on previously unaddressed aspects of corruption and tax evasion. Such multidisciplinary and comprehensive approaches promise not only to mitigate the limitations of current research but also to advance theoretical and practical knowledge in the field, guiding effective policy development and implementation.

In conclusion, this study offers a comprehensive analysis of the factors influencing tax evasion and revenue leakage in Somali customs, highlighting the critical role of systemic, governance, and personal dynamics. Through its theoretical contributions, policy implications, and practical applications, this research underscores the need for a holistic approach to addressing these challenges. By fostering an environment of transparency, accountability, and alignment of incentives, it is possible to mitigate the adverse effects of corruption and tax evasion, paving the way for economic stability and effective governance in Somalia and similar contexts.

## Supporting information

**S1 Dataset. My final dataset.**
(XLSX)

## Author Contributions

**Conceptualization:** Mohamed Ibrahim Nor.

**Data curation:** Mohamed Ibrahim Nor.

**Formal analysis:** Mohamed Ibrahim Nor.

**Funding acquisition:** Abdinur Ali Mohamed.

**Investigation:** Mohamed Ibrahim Nor.

**Methodology:** Mohamed Ibrahim Nor.

**Project administration:** Mohamed Ibrahim Nor.

**Resources:** Abdinur Ali Mohamed.

**Software:** Mohamed Ibrahim Nor.

**Supervision:** Mohamed Ibrahim Nor.

**Visualization:** Abdinur Ali Mohamed.

**Writing – original draft:** Mohamed Ibrahim Nor.

**Writing – review & editing:** Abdinur Ali Mohamed.

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
