## [Decision Letter · Decision Letter 0]

19 Feb 2024

PONE-D-24-02107Investigating the Dynamics of Tax Evasion and Corruption in Somali Customs: A Dual-Model ApproachPLOS ONE

Dear Dr. Nor,

Thank you for submitting your manuscript to PLOS ONE. After careful consideration, we feel that it has merit but does not fully meet PLOS ONE’s publication criteria as it currently stands. Therefore, we invite you to submit a revised version of the manuscript that addresses the points raised during the review process. SpecificallyAddress issues raised about your modeling approachAddress all comments raised by the reviewers

We look forward to receiving your revised manuscript.

Kind regards,

Stephen Esaku

Academic Editor

PLOS ONE

Journal Requirements:

"We express our profound gratitude to SIMAD University's Center for Research and Development for their generous funding and invaluable technical support throughout the course of this study. Their commitment to fostering research excellence and providing resources has been instrumental in the successful realization of our project. The insights and expertise offered by the team at the Center have significantly contributed to the depth and quality of our work. We are deeply appreciative of their unwavering support and belief in the potential of our study."

4. In the online submission form, you indicated that [Data will be available upon requet.]. 

5. Please ensure that you include a title page within your main document. You should list all authors and all affiliations as per our author instructions and clearly indicate the corresponding author.

6. We note that Figure 1 in your submission contain copyrighted image. All PLOS content is published under the Creative Commons Attribution License (CC BY 4.0), which means that the manuscript, images, and Supporting Information files will be freely available online, and any third party is permitted to access, download, copy, distribute, and use these materials in any way, even commercially, with proper attribution. For more information, see our copyright guidelines: http://journals.plos.org/plosone/s/licenses-and-copyright.

Reviewers' comments:

Reviewer's Responses to Questions

**Comments to the Author**

1. Is the manuscript technically sound, and do the data support the conclusions?

Reviewer #1: Yes

Reviewer #2: Partly

Reviewer #3: No

2. Has the statistical analysis been performed appropriately and rigorously? 

Reviewer #1: Yes

Reviewer #2: No

Reviewer #3: No

3. Have the authors made all data underlying the findings in their manuscript fully available?

Reviewer #1: Yes

Reviewer #2: No

Reviewer #3: No

4. Is the manuscript presented in an intelligible fashion and written in standard English?

Reviewer #1: Yes

Reviewer #2: No

Reviewer #3: Yes

5. Review Comments to the Author

Reviewer #1: 1. Based on the theories of principal-agent, information asymmetry and rent-seeking, this paper collects data from the government, the private sector and the social sector, and examines the causal relationship between corruption and tax evasion in the national context. The theory is solid and rich, the data are comprehensive and detailed, and the topic selection is valuable.

2. The two models of "the economic impact of tax evasion and corruption" and "the social and political impact of tax evasion and corruption" are explored. The two models are not completely separated but have a certain correlation, and further discussion on this part may be more complete.

3. The basis for describing the measurement method of each indicator is not presented and is not convincing enough.

4. In the quantitative analysis part, reliability analysis and heteroscedasticity test were carried out on the data, but multicollinearity test was lacking to avoid the strong correlation between the respective variables

The results are distorted, and the empirical analysis continues to be revised.

5.Whether this survey data in August 2023 is sufficiently representative of corruption and tax evasion in the country's society, and whether the results of the study vary according to the circumstances of a particular period, so robustness is necessary

6.Some fresh related papers can be asded, eg:

Sun H., Edziah B K., Sun C., Kporsu A K., 2021. Institutional quality and its spatial spillover effects on energy efficiency，Socio-Economic Planning Sciences, 101023. https://doi.org/10.1016/j.seps.2021.101023.

Reviewer #2: This is an important topic for Somalia. However, I found that too many jargons used in the analysis and it is very difficult to follow the story. I recommend the author to submit the paper to professional proof read service and submit it again.

Reviewer #3: I like quantitive approach and the PA/RS as a combined concept, PA indicates that A's have the power to act against the P (enabling corruption practices) and RS model discusses the incentive structure to choose corruption. Separating them into different regression equations seems not appropriate, but combining them into a single model that contains the "systemic opportunity structure" and and "personal incentive structure" seems a nice explanation for perceived tax evasion. This could also be nicely lined up with other studies on tax compliance that apply the slippery slope model, where the coercive power could facilitate the opportunity structure and the persuasive power the personal incentive structure.

Unfortunately this also leads to my biggest criticism: Your dependent variable does not measure tax evasion, but the individually perceived prevalence of tax evasion. Also the measure is drawn from the same survey that is used to identify your independent variables. So what your model actually measures is the extent in which the perception of the surveyed individuals on tax evasion is coherent with their perception of the system of policies. This would probably explain the high R-squared. (To ensure the high R2 is not caused by overfitting, perform some robustness checks, e.g. by randomly eliminating 15-20% of the observations and repeat this process, then compare the outcomes).

Technically the model seems to be not appropriate. The survey questions are manually grouped into "latent variables" but they are not independent, the same question must influence the answer to other questions. For example, CFTEC: weak regulatory enforcement should be directly associated with PIS: strengthening enforcement and penalties and ELRF: enforcement of penalties and sanctions. The two linear regression models do not take such interactions into account. A structural equation model (SEM) would therefore be more appropriate (ignoring the non-independent measure of tax evasion aforementioned).

Not all is lost, the survey is interesting and performing a exploratory (or possibly confirmatory) factor analysis to assess the "emergence" of the latent variables and explain them in the PA/RS setting would be useful and possibly generalizable. This could provide a framework in which the author can explain how counter TE policies could be designed to become more effective. Please be advised to add information on the role of the respondent as to place the responses properly (e.g. does the person work for the customs, is it a business person, or is the respondent a "victim" of TE practices). This could also contribute to a better explanation of the subjective view of the system (policy makers might think they do the right thing because they don't perceive the thing right). Changing the paper's scope to explain the perception of TE and find what determines such views can be very relevant for future policy design (so perform CFA) and could be publishable. Since you have the data and the framework, I did not reject the paper fully.

A few minor comments: your text has a lot of repetition which must be reduced. For example, ever sentence in section 2.2.2. claims the same argument: RS is the individual enrichment without creating societal value. Also, even if the author is spot-on with the policy recommendations, there is no clear scientific argument presented in the paper that backs these recommendations. So the author should better relate the policy recommendation with the model outcomes and literature review.

Finally, provide the data into a public repository to meet the journal's standards. Also, since you are using survey data, you need to have an informed consent form the respondents (which I can't find here) and you need to have an ethics commission reviewing your research. The piece could also gain form some language editing and changes in the structure.

6. PLOS authors have the option to publish the peer review history of their article (what does this mean?). If published, this will include your full peer review and any attached files.

Reviewer #1: No

Reviewer #2: No

Reviewer #3: No

---

## [Author Response · Author response to Decision Letter 0]

27 Mar 2024

We have addressed the issues raised about our modeling approach, and addressed all comments raised by the three (3) reviewers. 

Thanks

---

## [Decision Letter · Decision Letter 1]

8 Apr 2024

PONE-D-24-02107R1Investigating the Dynamics of Tax Evasion and Revenue Leakage in Somali CustomsPLOS ONE

Dear Dr. Nor,

Thank you for submitting your manuscript to PLOS ONE. After careful consideration, we feel that it has merit but does not fully meet PLOS ONE’s publication criteria as it currently stands. Therefore, we invite you to submit a revised version of the manuscript that addresses the points raised during the review process.

SpecificallyAddress the minor comments raised by Reviewer 3

We look forward to receiving your revised manuscript.

Kind regards,

Stephen Esaku

Academic Editor

PLOS ONE

Journal Requirements:

Reviewers' comments:

Reviewer's Responses to Questions

**Comments to the Author**

1. If the authors have adequately addressed your comments raised in a previous round of review and you feel that this manuscript is now acceptable for publication, you may indicate that here to bypass the “Comments to the Author” section, enter your conflict of interest statement in the “Confidential to Editor” section, and submit your "Accept" recommendation.

Reviewer #1: All comments have been addressed

Reviewer #3: (No Response)

2. Is the manuscript technically sound, and do the data support the conclusions?

Reviewer #1: Yes

Reviewer #3: Partly

3. Has the statistical analysis been performed appropriately and rigorously? 

Reviewer #1: Yes

Reviewer #3: I Don't Know

4. Have the authors made all data underlying the findings in their manuscript fully available?

Reviewer #1: Yes

Reviewer #3: Yes

5. Is the manuscript presented in an intelligible fashion and written in standard English?

Reviewer #1: Yes

Reviewer #3: Yes

6. Review Comments to the Author

Reviewer #1: This study aims to investigate the dynamics of tax evasion and revenue leakage in the Somali customs framework, providing insights into the systemic opportunity structures, tax governance deficiencies, and personal incentive structures that facilitate these practices. The current modifications are relatively complete, with a reasonable structure and strong arguments. It is neccessary to provide the data into a public repository to meet the journal's standards. It is necessary to optimize the innovation points at present, and further optimization can be carried out.

The format of the entire text needs to be unified according to the needs of the publication, for example, some literature does not have a unified citation format.

It is also necessary to deepen the discussion in the final part of the paper.

Reviewer #3: First I would like to congratulate the authors with the extensive rework they have done on implementing a new statistical approach and implementing factor analysis for their measurement model. The factor analysis is supporting the authors' allocation of latent variable indicators strongly, so this part of the analysis is done well. Also the authors have answered all the reviewers remarks, which is a considerable effort, so congratulations for that work too.

Unfortunately I still have a few more remarks, textual (1 and 3) and one empirically, (2):

1. Yet implementing the SEM approach also resulted in stating the hypotheses, but I cannot clearly find the support for the hypothesis in the preceding paragraphs. For example, the first paragraph seems to indicate both H1 and H4, yet H4 is listed later in the text. Also, I cannot find a justification for H2 and H5. Also, while H6 seems logical, I cannot find clear support within the text. Perhaps all these hypotheses are mentioned, but to me they are not clear enough and I would like to see e.g. "(H2)" in the text after the claim.

2. I played around with the provided data, in order to understand the model better, but I cannot explain the output of the SEM as reported by the authors. I recommend adding a diagram showing the full model (preferably with coefficients and p-values), which makes it easier to fully understand the model. My results depict a different outcome where I cannot find support for H1, H2 and H5. Therefore, if possible, please provide the used code to enable reproducibility (there is no way for me to validate the quality of your statistical analysis, hence this is why I have answered not to be able to assess if the manuscript is technically sound as well as the statistical analysis (PLOS ONE review questions 2 and 3).

3. Two minor remarks are that I would like to see the survey structure (with the answering options) to give a clear picture of how respondents could answer their questions. Also, the terminology used in Figure 1 seems to be not related to the associated text, and therefore I would recommend renaming the objects in figure 1 accordingly.

7. PLOS authors have the option to publish the peer review history of their article (what does this mean?). If published, this will include your full peer review and any attached files.

Reviewer #1: No

Reviewer #3: No

---

## [Author Response · Author response to Decision Letter 1]

15 Apr 2024

Rebuttal letters that respond to the comments of the academic editor and reviewers are uploaded in the system.

---

## [Decision Letter · Decision Letter 2]

30 Apr 2024

Investigating the Dynamics of Tax Evasion and Revenue Leakage in Somali Customs

PONE-D-24-02107R2

Dear Dr. Nor,

We’re pleased to inform you that your manuscript has been judged scientifically suitable for publication and will be formally accepted for publication once it meets all outstanding technical requirements.

Kind regards,

Stephen Esaku

Academic Editor

PLOS ONE

Additional Editor Comments (optional):

Reviewers' comments:

Reviewer's Responses to Questions

**Comments to the Author**

1. If the authors have adequately addressed your comments raised in a previous round of review and you feel that this manuscript is now acceptable for publication, you may indicate that here to bypass the “Comments to the Author” section, enter your conflict of interest statement in the “Confidential to Editor” section, and submit your "Accept" recommendation.

Reviewer #4: All comments have been addressed

Reviewer #5: All comments have been addressed

2. Is the manuscript technically sound, and do the data support the conclusions?

Reviewer #4: Yes

Reviewer #5: Yes

3. Has the statistical analysis been performed appropriately and rigorously? 

Reviewer #4: Yes

Reviewer #5: Yes

4. Have the authors made all data underlying the findings in their manuscript fully available?

Reviewer #4: Yes

Reviewer #5: Yes

5. Is the manuscript presented in an intelligible fashion and written in standard English?

Reviewer #4: Yes

Reviewer #5: Yes

6. Review Comments to the Author

Reviewer #4: Review Report

Comments for the authors:

1. The paper has an interesting topic and it is well structured.

2. The abstract is clear and has all the requirements for understanding the paper topic and highlights.

3. The introduction section is well constructed and is a roadmap for the paper. It encompasses all objectives, scope, and research strategy.

4. The literature review has an extended framework and contains both an analytical part, a theoretical part, and an empirical part.

5. The research hypothesis and questions and well structured and developed.

6. The material and methods part encounters all the information required for the development of the research techniques, procedures, and data analysis.

7. Results and well explained and tackle all the econometric tools and data interpretation methods.

8. The conclusion’s part is a very interesting and analytical one. It encompasses also some policy recommendations and explains in detail the pros and cons of the study.

9. The references list has been carefully constructed and cited in the text by the authors.

Reviewer #5: Thank you for sending his paper for review

The authors have addressed many of the comments. However, they should consider the following before publication.

1. The findings can be improved in the abstract section.

2. The motivation of the paper is needed to improve. The contributions of the study are suggested to be written clearly. Finally, the theoretical support for the topic is missing.

3. Why only focus on Somali? Provide reasons and rationale.

4. Updating the literature review part is required. It is advised to add the following to the literature: https://doi.org/10.1108/JFRA-06-2022-0234

https://doi.org/10.1108/IJPPM-09-2022-0486

https://doi.org/10.1016/j.jretconser.2023.103301

https://doi.org/10.3390/su152316138

https://doi.org/10.1007/s11301-023-00319-7

https://doi.org/10.1080/23311975.2023.2190195

https://doi.org/10.1016/j.najef.2021.101574

5. Expand the implications and insights from the findings of the study in the final remarks section.

6. More support from the literature should be provided for your writing.

7. Proofread the manuscript, as there are many typos.

7. PLOS authors have the option to publish the peer review history of their article (what does this mean?). If published, this will include your full peer review and any attached files.

Reviewer #4: No

Reviewer #5: No

---

## [Editor Report · Acceptance letter]

11 May 2024

PONE-D-24-02107R2 

PLOS ONE

Dear Dr. Nor, 

I'm pleased to inform you that your manuscript has been deemed suitable for publication in PLOS ONE. Congratulations! Your manuscript is now being handed over to our production team.

Kind regards, 

on behalf of

Dr. Stephen Esaku 

Academic Editor

PLOS ONE